# EXPLORE OUTWORLD KNOWLEDGE IN LARGE LANGUAGE MODELS: A CASE STUDY IN POKEMON GAME

## ABSTRACT

Large language models (LLMs) show great power by gathering almost all knowledge in our human world. An appealing curiosity now arises regarding their adaption to a new world setting, e.g from fictions and films, one with disparate fundamental laws, which is much more challenging than transferring between domains of the same human world. This carries significant research potential for expanding AI to multiple universes in the future. This paper chooses POKÉMON as the target, a popular strategy game with a unique worldview. We introduce POKEMON-PY, a Python library that provides an interactive playground as in the pokemon world. Our analysis demonstrates that the outworld context can exacerbate knowledge distortions and logical flaws in today's LLMs, and this phenomenon has a significant negative impact. Based on POKEMON-PY, we propose *Self-Training with Self-Competition*, a novel self-supervised learning method to effectively adapt the model to a new or even unknown world setting, where the model is programmed to keep learning through self-competition, and ultimately grows into a superior individual. Our method achieves remarkable improvement to adapt LLaMA2-7b to two downstream tasks within the pokemon world.

## 1 INTRODUCTION

Large language models (LLMs) (Chung et al., 2022; Chowdhery et al., 2022; OpenAI, 2023) have demonstrated the remarkable ability of AI to navigate the textual wealth of the human world in a super high efficiency, thereby acquiring nearly all human knowledge and cognitive power that even outstrips human beings. An appealing curiosity ensues: *can AI be generalized to a new world?*

While LLMs have generalized well between different scenarios and domains that are part of our familiar human world, which are exactly they are pre-trained on, with additional learning techniques (e.g. RLHF and SFT) aiming to aligning them with. In this paper, we focus on the knowledge in a new world. A new world is one in which there are disparate fundamental laws or even beyond the boundaries of human existence. It can be a virtual universe, for instance, the wizard world in Harry Potter and the planet Pandora in Avatar; or even a purely unknown space. LLMs faces a challenging task of reshaping their knowledge framework and overturning commonsense. We call the knowledge within these new worlds beyond the human world, *outworld knowledge*. The endeavor to generalize AI to outworld knowledge carries significant research potential across diverse domains, including metaverse, cinematic productions, electronic games.

In this paper, we study a specific case, the pokemon world, derived from POKÉMON, a popular strategy game. As opposed to other games that have been well studied in the community (Abramson et al., 2020; Shen et al., 2021; Küttler et al., 2020; Fan et al., 2022), which stick close to the human world, POKÉMON has a unique worldview with overhead creatures and laws. To explore the pokemon world, individuals can meet a variety of magical creatures called pokemons, and train them to battle against other players.

In the first part of this paper, we probe the pokemon knowledge in state-of-the-art LLMs, which serves as a subset of outworld knowledge. Our observation is that while LLMs memorize some pieces of pokemon knowledge, they fall into severe self-contradiction in their logic when reasoning is required. This phenomenon can have a significant negative impact, incurring inaccurate and misleading responses in the context of a new world.

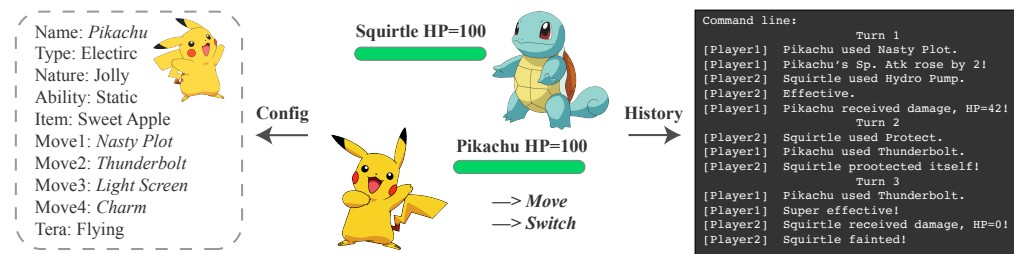

Figure 1: Illustration of POKEMON-PY, where two pokemons, *Pikachu* and *Squirtle* are battling. Left: Pokemons can be configured with a number of parameters. Right: The detailed battle history is provided in logs. One can play the battles by interaction in the command lines. It provides us with a steady source of data.

To adapt the language model to a new world setting, we propose *Self-Training with Self-Competition*. The model is programmed to compete against itself, weeping out flawed memories from the past and acquiring superior ones, and ultimately grows into a strong individual after a series of generations. The learning is self-supervised, driven by interaction with the environment, unlike text-based self-supervised learning (Devlin et al., 2019; Brown et al., 2020). To simulate the pokemon world, we introduce *Pokemon-Py*, an interactive environment built on Python.

We summarize our paper below:

• We present an interactive environment to stimulate pokemon battles (§ 2).

• We provide a qualitative analysis on the awareness of outworld knowledge in LLMs (§ 3).

• We propose a self-supervised learning method to adapt the model to an unknown world setting (§ 4).

• We design two downstream tasks to evaluate the outworld adaption of a model (§ 5).

## 2 COMPETITION IN THE POKEMON WORLD AND POKEMON-PY

In this paper, we focus on the competition in a new world. In the world of pokemons, this is mainly manifested as pokemon battles. This section presents the necessary background for pokemon battles as well as an overview of POKEMON-PY.

### 2.1 OVERVIEW OF POKEMON-PY

POKEMON-PY is an interactive Python library to simulate the battles in the world of pokemons. Compared to previous platform for online real-time battles[1], POKEMON-PY is offline with a large number of APIs, which facilitates researchers to develop promising algorithms.

We illustrate an overview of POKEMON-PY in Figure 1. In a single battle, there are two competitors (players) who manipulate a number of pokemons. Each player can send out one pokemon at a time onto the battlefield, with the rest as standbys. Each turn, the pokemon is allowed to take one action, unleashing a move or switching out to another pokemon. If one pokemon is defeated, another one from standbys should be sent out to the battle. Either of the competitors will win the battle when defeating all of the opponent pokemons.

To plan for next actions, the competitor is required to take a range of key elements into account. It is worth noting that explaining the following elements are necessary for our subsequent case study.

**Pokemon** Each pokemon is born with unique properties, e.g. types, attack and defense stats, hit points (HP or life points). In addition to predetermined elements, as shown in Figure 1, each pokemon can be customized in its nature, ability, item, moves, and tera type.

---

[1] https://pokemonshowdown.com/

**Move**  Each pokemon can learn four moves, each with a predetermined type, power, accuracy, and secondary effect. For instance, moves of a high power can cause greater damage to the opponent, while certain moves with no power yet possess a strong secondary effect, inducing benefits the user or producing undesirable conditions for the opponent.

**Item**  In a single battle, a pokemon is allowed to carry an item with it, serving various functions.

**Type**  Either a pokemon or a move belongs to a specific type or combination of two types and there are eighteen different types in the pokemon world. There is a matchup between types, which indicates the effectiveness of one type against another. For example, Water type is super effective to Fire type, while Fire type is resisted by Water type on the contrary. It is crucial to utilize the matchup between types while choosing moves to cause greater damage to the opponent.

**Tera**  During the battle, a player has once chance to switch a pokemon type to a different one by terastallizing (tera). Due to the change of type, the matchup of type effectiveness will also change.

## 2.2 BENCHMARK IN POKEMON-PY

It is hard to measure the competitiveness of a player to play pokemon battles. POKEMON-PY provides a number of rule-based imaginary opponents to automate this process.

**Random Player**  Random Player equally selects a move from all available moves in each turn of the battle. It will not switch the pokemon until the current pokemon faints and then will uniformly select a standby pokemon.

**MaxDamage Player**  MaxDamage Player always selects the move that will cause the most damage to the opponent by precisely calculating its power and type effectiveness of each available move. It is much stronger than Random Player.

To facilitate the process of selecting proper pokemons for the opponent, POKEMON-PY includes a range of predefined pokemons collected from the Internet. So far, there are 200 popular pokemons with detailed configurations. To evaluate a player against Random or MaxDamage Player, both sides are allowed to uniformly select a number of pokemons from the predefined pokemon pool. This random battle will be repeated for multiple times to reduce the variance.

In addition, POKEMON-PY integrates a series of databases, covering the detailed information in the pokemon world, e.g. pokemons, moves, abilities, items, which can be accessed by APIs.

## 2.3 PROBLEM DEFINITION FOR A LANGUAGE MODEL PLAYER

A language model player learns to plan the next action for the pokemon based on the context of the battle in the form of text.

We denote the language model parameters as $\theta$ and the corresponding classifier as $p_\theta(\cdot)$. Given a pokemon on the battlefield, we denote all its features as $\mathcal{P}_0$, including its name, hit points, types, item, available moves, and tera type. Similarly, we denote the opponent pokemon as $\mathcal{Q}_0$. The difference is that only the name and hit points are observable from $\mathcal{Q}_0$, while all the other features are hidden from the model. We denote the standby pokemons as $\mathcal{P}_1, \cdots, \mathcal{P}_{k-1}$, where $k$ refers to the number of available pokemons. The battle history is composed of a set of natural language sentences provided by POKEMON-PY, which we denote as $\mathcal{L}$.

Hence, the language model seeks to solve the probability:
$$p_\theta(\mathcal{Y}|\mathcal{P}_0, \mathcal{Q}_0, \mathcal{P}_1, \cdots, \mathcal{P}_{k-1}, \mathcal{L}) \tag{1}$$
where $\mathcal{Y}$ represents the action. For each turn in the battle, three actions are allowed:

• `move`: the model chooses an available move;

• `switch`: the model switches out the current pokemon and switches in another standby pokemon;

• `tera&move`: the model chooses an available move and in the meantime terastallizes the current pokemon before using the move.

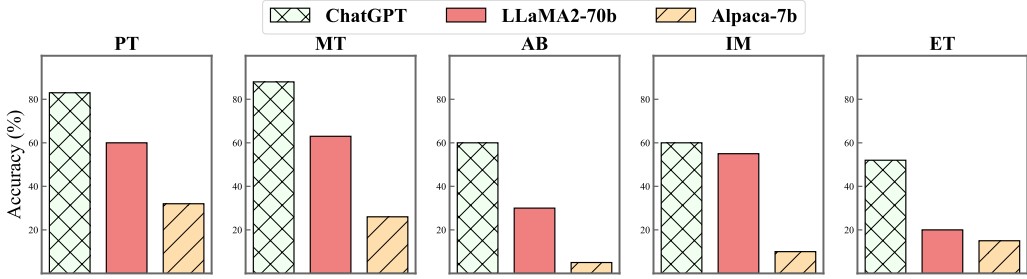

Figure 2: Accuracy in answering factual questions on various concepts.

# 3 ANALYSIS OF OUTWORLD KNOWLEDGE ON LLMS

In this section, we report empirical analysis on the outworld knowledge in state-of-the-art large language models (LLMs). We probe into their awareness of knowledge as well as capability to use it for logical reasoning.

We choose three LLMs for the experiment:

• ChatGPT (OpenAI, 2023): the strongest LLM to follow various human instructions;

• LLaMA2-70b (Touvron et al., 2023): one of the strongest open-source foundation language models pre-trained on a mix of public data sources;

• Alpaca-7b (Taori et al., 2023): a fine-tuned model to follow instructions based on LLaMA.

## 3.1 KNOWLEDGE

We select five primary concepts in the pokemon world and generate a set of factual questions on them, including pokemon types (PT), move types (MT), abilities (AB), items (IM), and type effectiveness (ET). Concretely, we select 100 questions for PT, MT, and ET respectively, with rule-based evaluation, and 20 questions for AB and IM, with human evaluation for them.

For all questions, we prompt the model with "In the pokemon world" to limit the scope of its responses. For example, we ask the model "In the pokemon world, what type is Pikachu?", "In the pokemon world, what is the effectiveness of Water type against Dragon type?". Besides, we follow the predefined instruction formats to improve the quality of responses for LLaMA2 and Alpaca.

We report the accuracy of the LLMs in answering various questions in Figure 2. We find that ChatGPT achieves the best accuracy, particularly in answering PT and MT, achieving an accuracy score over 80%. In contrast, the other two open-source models do not perform well. LLaMA2 achieves the accuracy score over 50% only on PT and MT, and shows limited awareness of more complex concepts like abilities and items. Alpaca performs poorly on all questions, suggesting that it is almost unaware of the pokemon knowledge.

Our finding indicates that LLMs retain part of the pokemon knowledge since they have been exposed to relevant corpus during the training process, while smaller LLMs are almost unaware of that knowledge.

## 3.2 LOGIC

Secondly, we study whether LLMs can utilize the outworld knowledge for reasoning. We construct a set of reasoning-required questions by asking the model to suggest a move for a pokemon against an opponent. The model is expected to offer a reasonable move from the four choices based on the properties of the opponent pokemon. For example, we prompt it with "In the pokemon battle, suggest a move for Pikachu against Chien-Pao, from the following moves.". Besides, we use the "Let's think step by step" prompt to encourage the model to offer its detailed thoughts (Wei et al., 2022; Zhang et al., 2022b; Yao et al., 2023).

Table 1: Representative cases when ChatGPT is requested to answer a factual question and to make reasoning, in which the knowledge in different responses contradicts itself.

| |
|---|
| *[Question]:* What is the effectiveness of Water type to Dragon type? |
| *[ChatGPT]:* Water type moves are not effective against Dragon type pokemons. (✓) |
| *[Question]:* Given an opponent pokemon, suggest a move for... Let's think step by step. |
| *[ChatGPT]:* Water type Hydro Pump is super effective to Dragon type Dragonite due to type weakness. (×) |

Table 2: Accuracy in answering factual and reasoning-required questions.

| | Fact % | Reason % | Reason (CoT) % | CoT % |
|---|---|---|---|---|
| ChatGPT | **72.6** | 32 | 39 | $18_{\downarrow 54.6}$ |
| LLaMA2-70b | 47.1 | 24 | - | - |
| Alpaca-7b | **22.4** | 18 | 22 | $6_{\downarrow 16.4}$ |

As it turns out, there are serious logical flaws even in the strongest ChatGPT. We showcase an example in Table 1 and we can see contradictory answers given by ChatGPT to the two questions. Specifically, it is aware of the fact that "Water type moves are not effective to Dragon type", while this is distorted in its reasoning procedure. It suggests that while the model retains a certain level of pokemon knowledge, it falters to harness it for reasoning, or may even suffers from forgetfulness.

Below, we provide quantitative results in Table 2 where LLMs answer 100 reasoning-required questions. We also report the overall accuracy in answering factual questions from previous Figure 2 for comparison. We access the correctness of the answers by human evaluation and report the following metrics:

• *Fact %*: accuracy in answering a given factual question;

• *Reason %*: accuracy in answering a reasoning-required question;

• *Reason (CoT) %*: accuracy in answering a reasoning-required question with the chain-of-thought prompt (LLaMA2 is not fine-tuned for CoT);

• *CoT %*: accuracy of the reasoning procedure (if any of the statements in it is wrong, we label the entire thought as wrong).

We find that while ChatGPT is very likely to give the correct answers to factual questions, it performs much worse on reasoning-required questions even with the assistance of CoT. However, the reasoning procedures provided by LLMs are filled with even more flaws, with ChatGPT and Alpaca only achieving an accuracy score of 18% and 6%, which drops significantly compared to factual questions. It is worth noting that the knowledge required for reasoning greatly overlaps with answers for factual questions.

This phenomenon is akin to the hallucination (McKenna et al., 2023; Agrawal et al., 2023; Mündler et al., 2023), an emerging issue within LLMs that they lean to offer counterfactual contents in their responses. Our results indicate that the hallucination issue can be even more serious in the face of outworld knowledge. In Sec. 5, we show that interacting with the environment is helpful to alleviate the hallucinations in LLMs.

## 4 METHOD

From the previous section, we highlights an issue within LLMs - the insufficient adaption to rare or specialized outworld knowledge, as exemplified by the pokemon knowledge. This phenomenon poses a significant challenge for LLMs to effectively navigate and comprehend a new world setting. For a brand new world, however, the various settings in it can be unknown, and one might barely have prior knowledge for it. Consequently, it is impossible to improve the adaption of LLMs by feeding them with a large amount of annotated data.

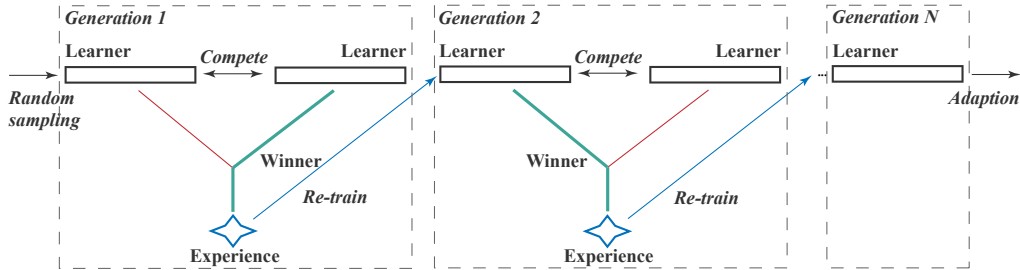

Figure 3: Training algorithm of *Self-Training with Self-Competition*.

## 4.1 SELF-TRAINING WITH SELF-COMPETITION

Though the model is unknown about the mechanism behind the world, it can learn through its own experiences and thus improves itself step-by-step. The experiences derive from interacting with the world environment, which completely adheres to the laws of this world. However, there are various forms of interaction in the world, while certain forms cannot provide useful signals.

Competition is the heart of evolution, providing individuals with useful signals that allow for growing into stronger ones. We propose to program the model to compete against itself, and the model in the later generations to learn the victorious experiences from the past. We call it *Self-Training with Self-Competition*, simply self-training in the following experiments.

Concretely, based on POKEMON-PY, which performs the duty of hosting a competition in the pokemon world, taking the form of two sides playing pokemon battles.

As depicted in Figure 3, the language model acts as a start learner from the beginning. We acquire the first batch of training data by random sampling, uniformly choosing one move in each turn, and use this data to train the initial model. The resultant model learns to make choices of moves in battles, even though its choices are random. We make a copy of this model and have the two of it play against each other for multiple times. The outcome of a battle is that one side of them wins. We ignore the rare case where there is a draw, which occurs in a very low probability. For each time of the battle, we trace and record the actions of the winner. We thus acquire a new batch of training data, which is stronger than the last, and retrain a new model using the new data (reinitialize the model). Similarly, we have two new models to play against each other and record the actions of the winner every time. We keep iterating over this process and each iteration is referred to a generation. After a series of generations, we eventually obtain the strongest model.

To avoid the learning process falling into a local optimum, we set a probability $\epsilon$ in which the model will choose a random move rather than its own decision. The choice of $\epsilon$ affects the convergence rate of the algorithm, and a larger one will make it slower. In our experiments, $\epsilon$ is set to 0.2 heuristically and the empirical results are in Appendix A.1.

Because POKÉMON is a well-known game, there are multiple language resources related to it, while we chose not to use them. We assume that the pokemon world is new or unknown to the learners. Therefore, our method can be generalize to other world settings.

## 4.2 TRAINING SETUP

We present setups for our experiments. For two competitors, we randomly assign them a pokemon chosen from the pool of predefined pokemons. The problem definition for a language model competitor is in Sec. 2. POKEMON-PY provides all battle information.

In each generation, we replicate the model and have them compete against each other for multiple times, and continuously collect data data from the winners til 5,000 samples. Each generation takes about 8 GPU hours. We train 50 generations of the model.

We choose LLaMA2-7b as the learner and adopt LoRA (Hu et al., 2022) to improve the training efficiency. The learning rate is chosen from {3e-4,8e-4} and batch size from {2,4} for each chip.

Table 3: Victory rate against two rule-based players over 100 times and 3 random seeds.

|  | Gen 1 | Gen 2 | Gen 3 | Gen 5 | Gen 10 | Gen 20 | Gen 30 | Gen 40 | Gen 50 |
|---|---|---|---|---|---|---|---|---|---|
| Random | 52.2 | 53.6 | 58.5 | 61.2 | 75.5 | 75.5 | 80.8 | 83.7 | 89.7 |
| MaxDamage | 11.8 | 18.0 | 23.0 | 32.7 | 40.0 | 51.0 | 67.6 | 73.2 | 72.0 |

The learner is trained to predict the move option in each turn through the next token prediction loss.

### 4.3 EMPIRICAL RESULT

We evaluate the performance of each generation by battling against Random Player and MaxDamage Player. Table 3 summarizes the victory rate of each model against two rule-based players for 100 battles. Initially, we observe that the LLaMA2 model performs on par with a random selector. By the third generation, it makes noticeable progress and starts to surpass Random Player. By the tenth generation, it makes a significant stride and achieves an impressive victory rate of 75.5% against Random Player. As it advances to the thirtieth generation and beyond, the model continues to boost, reaching a remarkable victory rate of 89.7% in the fiftieth generation.

MaxDamage Player is much strong than Random Player. It is nearly invincible from the start. However, we observe a big boost in the fifth generation, where the victory rate comes to 32.7%. The truth behind is that the model learns the type effectiveness between different types. It suggests that superficial mapping relations are easier to learn for LLMs, while deeper strategies are difficult to explore. It experiences a flat growth over the next ten generations. By the twentieth generation, its victory rate against MaxDamage Player just reaches 51%, while it has achieved a victory rate of 75.5% against Random Player. However, by the thirtieth generation, the victory rate surges to 67.6%, and after the fortieth generation, the model eventually achieves a victory rate over 70% against MaxDamage Player.

## 5 EVALUATION OF OUTWORLD ADAPTION

We have demonstrated that the model grows stronger through learning from self-competition. However, it is unclear that whether it has adapted to the new world. We notice that it is hard to directly measure the level of adaptation because there is no matching task. Hence, we design two downstream tasks and fine-tune the resultant model on them. Note that both tasks are derived from Sec. 3, but we make them simpler for automatic evaluation. The fine-tuning results indicate whether the model can better adapt to various specific tasks in the world of pokemons.

**Boolean Question-Answering** The first task is similar to factual question-answering. The difference is that the fine-tuned model can only respond *true* or *false* for the given question. Specifically, we sample 200 factual knowledge as the positive samples (answer true). On top of them, we conduct random substitution of characters to obtain the negative samples (answer false). The model is trained on these 400 samples with half positive and half negative samples. For test samples, we keep the positive training samples and conduct another random substitution of characters to obtain new negative samples. Since the positive samples are the same from training data to test data, we report the performance on them separately.

**Language Inference** The second task is more complicated, which requires the model to inference and offer the procedure. As illustrated in Table 4, we give the model a move and a pokemon. The first step for the model is to tell the types of them. The second step is to inference the type effectiveness between them based on the first result. We randomly combine different moves and pokemons by rules and get 200 training samples 100 test samples. The accuracy of this task is calculated by human evaluation. We notice that this task serves a similar purpose to measure LLMs' hallucinations.

We fine-tune two models, with original LLaMA2-7b weights and the pre-learned weights via self-training, and average the results over three random seeds. The results are summarized in Table 5, where we pick the checkpoints in four generations for comparison. We observe that the model demonstrates a substantial improvement after self-training in its proficiency for discerning facts and

Table 4: Examples of two downstream tasks.

| Boolean QA | |
|---|---|
| Instruction: Given a statement, tell it true or false. Input: *Pikachu* is Electric type. Response: true | Instruction: Given a statement, tell it true or false. Input: *Pikachu* is Ghost type. Response: false |

| Inference |
|---|
| Instruction: Given a move and a pokemon, inference the type effectiveness. Input: *Earthquake* and *Pikachu* Response: *Pikachu* is Electric type and *Earthquake* is Ground type. Ground type moves are effective to Electric type pokemons, so *Earthquake* is effective to *Pikachu*. |

Table 5: Fine-tuning results on downstream tasks with standard deviations over 3 seeds.

| | Boolean QA | Inference |
|---|---|---|
| LLaMA2 | 96.0 / 58.0$_{(1.14)}$ | 46.0$_{(1.63)}$ |
| LLaMA2 - *Self-Training (Gen 5)* | 96.0 / 64.0$_{(0.86)}$ | 46.0$_{(0.00)}$ |
| LLaMA2 - *Self-Training (Gen 20)* | 94.4 / 68.0$_{(0.90)}$ | 49.7$_{(1.25)}$ |
| LLaMA2 - *Self-Training (Gen 30)* | 97.9 / 74.5$_{(1.56)}$ | 55.0$_{(0.82)}$ |
| LLaMA2 - *Self-Training (Gen 50)* | **98.6 / 75.0**$_{(1.03)}$ | **60.0**$_{(2.90)}$ |

inference. On Boolean QA, specifically, the model does well in memorizing positive samples as well as recognizing negative ones after self-training. It substantiates our method, underscoring that the self-competition can effectively help the model to adapt to the pokemon world. In addition, we find that the easier boolean QA task is gained more quickly, while the inference task doesn't show a significant rise till generation 30. This also suggests that the gain derives from interaction with the environment, rather than the corpus itself, since models in all generations see similar corpus.

# 6 CASE STUDY

In this section, we take a closer look at what the model learns from self-training.

**Learning to use non-damage moves** As a start learner, the model turns out to learn the short-term impact of the moves, in terms of the damage that they can cause to the opponent immediately. As a result, the model lean to choose the move with a higher power and type effectiveness, similar as MaxDamage Player. However, this is still a naive strategy for a sophisticated player. From Table 6 (a), we can find that the model in generation 8 has learned to use *Will-O-Wisp* at the beginning of the battle. This move will not cause damage, in exchange for weakening the opponent, thus acquiring for long-term gain.

**Learning to terastallize** Terastallization is not taught to the model, since the model is initialized from randomly sampled training data. From Table 6 (b), however, we find that the model in the last generation learns the correct usage of terastallization, where the pokemon *Magnezone* terastallizes to Fairy type such that the move *Tera Blast* is super effective to the opponent.

In the above two cases, the model successfully learns more complex strategies and new concepts through learning from self-competition.

# 7 RELATED WORK

This paper explores the outworld knowledge in large language models (LLMs) (Chowdhery et al., 2022; OpenAI, 2023; Zhang et al., 2022a; Chung et al., 2022) in the hope of generalizing today's LLMs to multiple universes. We select the pokemon world as our target, derived from a popular strategy game. There are other games that have been studied in the AI community, e.g. Atari (Mnih et al., 2013), Minecraft (Fan et al., 2022), NetHack (Küttler et al., 2020), Werewolf (Xu et al., 2023), and beyond (Wang et al., 2019; Juliani et al., 2019; Abramson et al., 2020; Shen et al., 2021).

Table 6: Case study where the model is asked to offer a move suggestion given the battle context.

| (a) *[Input]* | |
|---|---|
| pokemon: *Skeledirge*
moves: *Slack Off*, *Torch Song*, *Hex*, *Will-O-Wisp*
opponent: *Kingambit*
history: | pokemon: *Skeledirge*
moves: *Slack Off*, *Torch Song*, *Hex*, *Will-O-Wisp*
opponent: *Kingambit*
history: *Kingambit received damage, HP=180/404...* |
| *[Generation 1 Response]:* | |
| *Torch Song*   [a move to cause damage] | *Torch Song*   [a move to cause damage] |
| *[Input]* | |
| pokemon: *Skeledirge*
moves: *Slack Off*, *Torch Song*, *Hex*, *Will-O-Wisp*
opponent: *Kingambit*
history: | pokemon: *Skeledirge*
moves: *Slack Off*, *Torch Song*, *Hex*, *Will-O-Wisp*
opponent: *Kingambit*
history: *Kingambit fell into Burn...* |
| *[Generation 8 Response]:* | |
| *Will-O-Wisp*   [a move to weaken the opponent] | *Torch Song*   [a move to weaken the opponent] |
| (b) *[Input]* | |
| pokemon: *Magnezone*
moves: *Thunder*, *Flash Cannon*, *Rest*, *Tera Blast*
opponent: *Roaring Moon*
history: | pokemon: *Magnezone*
moves: *Thunder*, *Flash Cannon*, *Rest*, *Tera Blast*
opponent: *Roaring Moon*
history: |
| *[Generation 10 Response]:* | *[Generation 50 Response]:* |
| *Thunder*   [not effective] | *Tera Blast, tera Fairy*   [super effective] |

However, we do not focus on playing games by AI, but rather creating AI for a new world. Our work is different from learning an AI agent or interacting with LLMs for planning (Park et al., 2023; Wang et al., 2023; Zhu et al., 2023). Instead, we focus on world-level understanding rather than policies for specific objectives. It is a high-level perspective of unsupervised domain adaption (Ganin & Lempitsky, 2015; Ramponi & Plank, 2020).

We analyze LLM's awareness of the outworld knowledge, and our findings are relevant to hallucinations (McKenna et al., 2023; Agrawal et al., 2023; Mündler et al., 2023) and logical robustness (Zhou et al., 2021; Sanyal et al., 2022), where the model is weak against offering self-contradictory responses. Our method is shown to alleviate hallucinations, and thus is promising to general LLMs.

In the text domain, self-supervised learning methods typically generate learnable input text from unlabeled corpus, e.g. masked language modeling (Devlin et al., 2019), contrastive learning (Gao et al., 2021), text ennoising (Lewis et al., 2020; Wu et al., 2022). In contrast, the self-supervised signals in our method come from competition in the environment, which weeds out inferior samples and produces strong ones. We believe its idea is akin to genetic algorithms (Mitchell, 1998), with the specific fitness function referring to the competitiveness for pokemon battles.

The mechanism of self-play has been discussed in other contexts in previous work (Vinyals et al., 2019). For instance in emergent communication (Lowe et al., 2020), the authors propose to have two agents cooperate with each other to boost the adaption to a new language, while our focus is self-competition during the process of self-play, which is the most general way to screen out for better data or individuals in nature. This also means our method can be totally unsupervised.

# 8   CONCLUSION

This paper presents an empirical case study in the pokemon world for outworld generalization of LLMs, and shows that existing LLMs are poorly skilled at outworld knowledge. A self-supervised learning method based on competition is proposed, and shows its effectiveness on two downstream tasks. In addition, a Python environment to simulate the pokemon world is introduced.

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

# A APPENDIX

## A.1 SELECTION OF $\epsilon$

Table 7: Results with different values of $\epsilon$.

|  | Gen 1 | Gen 2 | Gen 3 | Gen 5 | Gen 10 | Gen 20 | Gen 30 |
|---|---|---|---|---|---|---|---|
| $\epsilon=0$ | 51.5 | 52.7 | 56.2 | 59.5 | 68.8 | 62.0 | 68.5 |
| $\epsilon=0.2$ | 52.2 | 53.6 | 58.5 | 61.2 | 75.5 | 75.5 | 80.8 |
| $\epsilon=0.5$ | 51.9 | 51.4 | 52.3 | 52.1 | 55.9 | 58.8 | 62.6 |

We see that $\epsilon$ plays an important role for the eventual performance, while the model converges much slower when $\epsilon=0.5$, since the randomness is too high.

