# OpenReview forum: "Explore Outworld Knowledge in Large Language Models: A Case Study in Pokemon Game"
_ICLR.cc/2024/Conference — Submitted to ICLR 2024_

### Official Review · Reviewer_gZuL · 2023-10-31

**Soundness:** 3 good
**Presentation:** 3 good
**Contribution:** 2 fair
**Rating:** 6
**Confidence:** 3

**Summary:**

The paper explores the adaptability of LLMs to outworld knowledge by focusing on the Pokemon world, which is distinct from our human reality. The authors introduce an interactive Python environment called Pokemon-Py to simulate the Pokemon world and analyze the model's performance. They find that while LLMs can memorize facts about the Pokémon world, they struggle with logical inconsistencies. To address this, they propose a new self-supervised learning method called "Self-Training with Self-Competition," which allows the model to better adapt to new and unknown settings.

**Strengths:**

1. Innovative focus on the ability of LLMs to adapt to outworld knowledge, a largely unexplored area.
2. Introduction of a specific Python environment (Pokemon-Py) for simulating and testing the model in a new context.
3. Proposes a novel self-supervised learning method that shows promise in improving the model's adaptability to new worlds.

**Weaknesses:**

1. The paper focuses solely on the Pokemon world, so it's unclear how generalizable the findings are to other outworld settings.
2. The work may not delve deep enough into the specifics of why LLMs struggle with logical reasoning in new worlds.
3. There is a lack of discussion on potential real-world applications beyond gaming scenarios.

**Questions:**

1. Could the authors give some intuitions on why LLMs face logical inconsistencies when reasoning about the Pokemon world?
2. How generalizable is the "Self-Training with Self-Competition" method to other outworld or fictional settings?
3. Does the model's self-competition lead to any form of model destabilization or other unexpected behaviors?

---

> ### Author Response · Authors · 2023-11-17
>
> Thank you for your valuable review.
>
> **W2&Q1** why LLMs face logical inconsistencies about the Pokemon world
>
> As we have discussed in the paper, LLMs will encounter logical inconsistencies even facing general questions in the human world, e.g. hallucinations, even in simple Boolean logic [1]. When transferring to the Pokemon world, this issue is even amplified because:
>
> * The vast majority of pokemon data on the web is like wikipedia, describing the basic attributes of each Pokemon, **while the available data that relates reasoning is very minimal.**
>
> * **Most high-quality data is on videos**, which is hard to collect. We notice that there are methods that allow the model to learn from videos, which is an interesting future work for us.
>
> **W1&Q2** how our method generalizes to other new worlds
>
> Thank you for the question. We admit that in the paper we only discussed the outworld of the Pokemon world and do not verify our findings in other new world settings. These are highly interesting to discover, and we will leave these new things for future work, thank you.
>
> **W3** discussion of the real-world applications beyond game scenarios
>
> It will be fascinating to customize an LLM for different outworlds, to create a sense of unprecedented immersion. This is applicable for the movie industry, theme parks, beyond games. The ultimate application may be the metaverse, which can be connected to each of our daily life.
>
> **Q3** does self-competition lead to any unexpected behaviors
>
> Thank you for your question. We have not found any unexpected outcomes of the model after self-training.
>
> [1] Hongqiu Wu, et al. Empower Nested Boolean Logic via Self-Supervised Curriculum Learning

---

> > ### Comment · Reviewer_gZuL · 2023-11-22
> >
> > Thank you to the authors for the clarification and the additional experiments. My opinion remains in favor of accepting the paper.

---

> > > ### Author Response · Authors · 2023-11-23
> > >
> > > Thank you for your reply.

---

### Official Review · Reviewer_hwhv · 2023-10-31

**Soundness:** 3 good
**Presentation:** 2 fair
**Contribution:** 3 good
**Rating:** 6
**Confidence:** 3

**Summary:**

The paper presents an exploration of how large language models (LLMs) can adapt to new world settings, such as the Pokemon world, which differs significantly from the human world. It introduces POKEMON-PY, a Python library that simulates the Pokemon world and provides an interactive environment for LLMs to learn. The paper also analyzes the awareness and reasoning of outworld knowledge in LLMs, highlighting severe distortions and contradictions. A self-supervised learning method based on self-competition is proposed, where the LLM improves itself by playing Pokemon battles against itself. The paper concludes by evaluating the outworld adaptation of the LLM on two downstream tasks, demonstrating significant improvement after self-training.

**Strengths:**

1. This paper introduces a new setting called outworld knowledge, which might be a new direction for future (LLM) agents.
2. This paper proposes a new environment Pokemon-Py. However, I think the environment is somewhat simplistic as it only involves combat operations.
3. This paper shows that under the guidance of the winning rate, LLMs can learn new outworld knowledge without manual labeling.

**Weaknesses:**

1. I suggest the authors to give a definition or formulation on "outworld knowledge". And if this field has been previously researched before, the author should cite it. This can enable readers to more accurately get the problem the author wants to address, rather than having a vague concept.
2. The method part seems to lack some innovativeness. Similar approaches might have been proposed in prior work, such as AlphaStar. Furthermore, it appears to share some similarities with RLHF or DPO, except that the annotator has transitioned from a human to the environment. However, I don't deny that this method can be helpful for transferring LLM to a new environment.
3. I think this environment is a good contribution. Researchers can generate some entirely counterintuitive operations by configuring the environment's basic settings, such as making fire counteract water (which is not Pokemon but a completely unfamiliar environment that needs exploration). Therefore, I hope the author can provide a more detailed description of the extensions that this environment can offer and how they may be applied in research scenarios.

**Questions:**

1. Can the authors add a baseline "playing against humans"? This is a much stronger baseline than Random and MaxDamage.

2. I believe that the main contribution of this paper is not the introduction of a learning approach to achieve a higher winning rate, but rather the demonstration of some form of transferability within LLM (because given the current context, game theory or search may potentially achieve better results than LLM.) Could the author discuss this issue?

---

> ### Author Response · Authors · 2023-11-17
>
> Thank you for your valuable review.
>
> **W1** definition of outworld knowledge
>
> Thank you for your suggestion. We have defined “outworld knowledge” in the second paragraph of Section 1.
> We notice that there may be a lack of discussion of the relevance of our defined outworld knowledge to previous studies. We will work on this point carefully in the new version. Thank you.
>
> **However, in the context of emerging LLMs, we are the first to discuss the model adaption to new world settings, to the best of our knowledge.**
>
> **W2** lack of innovativeness of the method
>
> First, we notice that our method is simple yet effective, and there are similar variants proposed in previous papers, but for different purposes, as suggested by Reviewer 5UvA.
>
> **However, from a high-level perspective, the concept of environment feedback we present in the paper is more general than human feedback,** which is embodied in the human environment. This bridges the gap to transfer the model to other new worlds.
>
> Another merit is that **our method does not necessitate a reward model** since the environment can be used all the time.
>
> **W3** detailed description of the extensions of Pokemon-Py
>
> Thank you for your advice. **Based on our implementation, it is easy to extend it to present diverse environments in the pokemon world.** For example, to change the type effectiveness, one simply needs to modify a number in the type configuration file. We will detail how Pokemon-Py can be applied in the appendix. Thank you.
>
> **Q1** baseline against humans
>
> Yes, we agree that playing against humans is a stronger baseline than MaxDamage. However, please allow us to clarify that **the goal of self-training is to learn a better initialization to adapt to a new world**, which means to perform better on new world downstream tasks.
>
> Contrarily, to compete with an experienced human player (e.g. some tricks to confuse the opponent), this outworld knowledge is far from enough. **We argue that the performance against MaxDamage player is a more intuitive reflection of outworld knowledge acquisition.**
>
> Hence, we respectfully argue that including human players is not a mandatory reference to prove the effectiveness of our method. Sure, we are willing to report it in our new version, thank you.
>
> **Q2** whether our paper focuses on achieving a higher winning rate or insight on transferability
>
> Yes, we agree with your opinion. As we clarify above, **forging a sophisticated LLM agent to compete against humans is neither the centric nor contribution of our paper.** From our empirical experience, LLMs even GPT4 cannot achieve human-like performances in playing games, though there is a chance if training a policy on them.
>
> In our paper, we first unveil the powerlessness of LLMs on outworld knowledge. We then propose to utilize the nature of competition to generate strong data to transfer the model. The performance against MaxDamage player and on two downstream tasks demonstrate the better adaption of the trained model.

---

> > ### Comment · Reviewer_hwhv · 2023-11-22
> >
> > Thanks for your rebuttal.
> >
> > **W1**: I know the novelty of the "outworld" setting but I want to see a **formal** definition if possible since paragraph 2 in the introduction is somewhat ambiguous. I think something like this would be good: "The distribution here represents the in-world knowledge which means that LLMs have been pre-trained on it. Here is the human preference distribution and SFT and RLHF try to align LLMs with this distribution. And here is the out-world knowledge which represents the knowledge in the new world, and this is exactly the problem that we want to estimate and solve."
> >
> > **Q1**: Yes I agree that your contribution is not focused on beating humans. But I think a human baseline can make this paper more complete especially for a novel problem (outworld knowledge) because it shows how far the methods are from human-level intelligence.
> >
> > I have no problems with the other parts. Overall I keep my score and vote for acceptance.

---

> > > ### Author Response · Authors · 2023-11-22
> > >
> > > Thank you for your reply. We agree with your suggestion in W1 and will follow that to improve our paper. In addition, we will further discuss the human baseline in our new version. Thank you!

---

### Official Review · Reviewer_5UvA · 2023-11-01

**Soundness:** 3 good
**Presentation:** 3 good
**Contribution:** 3 good
**Rating:** 8
**Confidence:** 4

**Summary:**

As LLMs show incredible capacity to gather knowledge about the human world through their supervised pre-training, the question of whether they are able to gather knowledge about completely other (such as fictional) world arises.

This paper proposes to investigate the ability from AI to be transferred from human world to other worlds, such as fictional worlds like the game of POKEMON, and refer to this as outworld adaptation/generalization.

In doing so, this paper contributes:

1. 'outworld knowledge awareness' : a baseline evaluation of the pokemon-related outworld knowledge present in ChatGPT, LLaMA2-7b and Alpaca-7b, in terms of factual and reasoning-requiring Q&As, showing poor result on the reasoning-required Q&As and therefore goading us to the conclusion that the outworld laws of the pokemon world are confusing state-of-the-art LLMs.
Note that the reasoning-required questions are based on accurate move selection for a pokemon in battle with an opponent.

2. POKEMON-Py : a python library enabling using POKEMON battles as interactive playground for text-based state-of-the-art AI systems: text-based observations and instructions are provided to simulate a two-players pokemon trainer battle.

3. Self-Training with Self-Competition : a novel self-supervised learning method to adapt pre-trained LLMs to new world settings, thus enabling growth.

Regarding evaluation of the outworld adaptation, the paper proposes to instrumentalise two downstream tasks performance as a measure of outworld adaptation.

Experimental evidences shows that LLaMA2-7b can be effectively adapted to the POKEMON world.

**Strengths:**

## Originality:

As far as I know, outworld adaptation/generalization is a novel problem and the papers proposes a new environment in POKEMON-PY.

## Quality:

The paper addresses a valuable problem and proposes interesting experiments.
Reproducibility seems fairly high.

## Clarity:

Figures and explanations are fairly thorough.

## Significance:

Valuable problem to address and the proposed resource is hoped to be useful to the community.

**Weaknesses:**

## Novelty:

As far as I know, outworld adaptation/generalization is a novel problem and the papers proposes a new environment in POKEMON-PY.

While the proposed self-supervised learning method to address this new outworld adaption problem is new, in the context of LLMs and outworld adaptation, it is actually very similar to Supervised Self-Play (S2P) in its 'scheduled' variant from [1], which was proposed in the context of Emergent Communication ; which can be indeed thought of as an outworld adaptation where the outworld is the natural language and the pre-trained world is that of the emergent language.

In more details, scheduled S2P is stage-wise identical to the algorithm proposed here.
Where a difference can be seen is in 2 points:

1. firstly, the dataset that is used during the supervised learning stage is static in scheduled S2P because there is no RL-like environment to sample new data from.
2. secondly, the self-play task is cooperative in scheduled S2P, it is a referential game, while the proposed algorithm here is using a competitive task.

I find that those two discrepancies bring enough incremental innovation to make the paper worthy of publication provided that a proper discussion about the similarities and differences with scheduled S2P is included to the manuscript.


[1] : Lowe, Ryan, et al. "On the interaction between supervision and self-play in emergent communication." International Conference on Learning Representations. 2019.

## Quality:

### Unsupported Claim:
In Section 4.1, the following claim is made without evidence:
`The choice of ε affects the convergence rate of the algorithm, and a larger one will make it slower.'
Please provide citation of a similar phenomenon in previous literature or supporting evidence such as a table or a graph showing, respectively, either the final performance or the learning curves of different runs within the different hyperparameter settings of $\epsilon$.

### Statistical Significance:

While Table 3 and 5 show very interesting progressions, I think that the experiments both lack in terms of statistical significance:

1. It seems that only one random seeded run has been performed, since there is no mention of how many random seeds have been run. I would expect at least 3 differently-seeded runs to be reported on, if possible, please?

2. Then, following usage of multiple randomly-seeded runs, I would expect the reported statistics to be comprised of at least mean (or median) and standard deviation (or variance).

I would expect to see in Section 4.3 the details about the different random seeds, if it is solely an omission.

### Experimental design :

Section 5 highlights boolean Q&A and Language Inference as the two downstream task to evaluate outworld adaptation, following the self-supervised learning stage.

I am surprise that the experiments in Section 3 are not being repeated following the self-supervised learning stage in order to evaluate the outworld adaptation.
Could you explain why is that?


## Clarity:

### Need for an algorithm?

Section 4.1 details the proposed self-supervised learning method, but I find it difficult to follow.
I think that the addition of formal notation to refer to each model and the training operations that they go through would help increase the clarity, maybe?

Or, if space permits it, I think it would be helpful to provide an actual algorithm environment to refer to, on top of the information provided in Figure 3, maybe?


### Hyperparameter choice:

Section 4.3 highlights batch size and learning rate hyperparameter values but it is unclear how those are used:
Has there been multiple runs of the experiment with each combination of the hyperparameter values?

### Section 5 Test-Train Split Issue ?

The boolean Q&A fine-tuning and testing experiment is unclear with regards to the train-test split employed: from my understanding, the positive testing samples are the same as the positive training samples?
If so, then this is a critical issue in terms of the validity of the experiment.

## Significance:

In the current state, both (i) the gap in related work comparison, and (ii) the missing statitics to truely evaluate the significance of the results make the significance of the paper difficult to evaluate.

For now, I can only vouch for a minimal significance.
I am hoping to be able to increase my appreciation throughout the rest of the review process and hope to increase my scores accordingly.

**Questions:**

Please see Weaknesses above.


# AFTER REBUTTAL :

Following the rebuttal, I am very satisfied with the answers and ~~(proposed)~~implemented changes.
If accepted, I am hoping the authors will carry on in the current direction of their implemented changes, and therefore I am increasing my score to ~~6~~ 8.

---

> ### Author Response · Authors · 2023-11-17
>
> Thank you for your careful and insightful review.
>
> **W1** discussion with relevant papers
>
> Thank you very much for suggesting a relevant paper. After reading this paper, we agree that there are similarities between our proposed self-training and self-play (S2P). We both have two models to play with each other, while the purposes are different.
> Our focus is self-competition during the process of self-play, **which in our opinion is the most general way to screen out for better data or individuals,** in nature. This also means our method is totally unsupervised.
> Instead, the purpose of S2P is self-cooperation, which largely relies on the supervised data, as discussed in the paper.
>
> Thank you for your suggestion. We will carefully discuss the relationships in our new version.
>
> **W2.1** the choice of $\epsilon$
>
> Thank you for your question. The idea of choosing $\epsilon$ is the same as \epsilon-greedy. According to our empirical experiments, it is useful. We show the results against Random player before Gen 30 below:
>
> |              | Gen 1 | Gen 2 | Gen 3 | Gen 5 | Gen 10 | Gen 20 | Gen 30 |
> |--------------|:--------:|:--------:|:--------:|:--------:|:--------:|:--------:|:--------:|
> |$ \epsilon=0$   | 51.5  | 52.7  | 56.2  | 59.5  | 68.8   | 62.0   | 68.5   |
> |$ \epsilon=0.2$ | 52.2  | 53.6  | 58.5  | 61.2  | 75.5   | 75.5   | 80.8   |
> | $\epsilon=0.5$ | 51.9  | 51.4  | 52.3  | 52.1  | 55.9   | 58.8   | 62.6   |
>
> We see that $\epsilon$ plays an important role for the eventual performance, while the $\epsilon=0.5$ case converges much slower, since the randomness is too high.
>
> Due to our multi-aspect experiments in the paper, we cannot discuss every setting in the paper with limited space. We will include these numbers in the appendix. Thank you.
>
> **W2.2** statistical significance
>
> Thank you for your reminder. In the paper, to reduce the variance, **we perform each battle play against Random and MaxDamage players with 100 times, which we have mentioned in the paper.**
>
> In addition, we clarify that **all experiments are ran under 3 random seeds (42, 0, 2023), and we report the average numbers.** Since each result are averaged from 300 battles, **the variance is very low** even in the early generations of self-training.
>
> The self-training process (for 50 generations) is only performed for once because of the great training cost.
>
> For the fine-tuning experiments on two downstream tasks, **we ran only for once in the paper,** and this is a drawback. **Hence, we rerun the experiments over 3 seeds** (42, 0, 2023). We report the new results as well as the standard deviation below:
>
> |        | Boolean QA  | Inference   |
> |--------|:-----------:|:-----------:|
> | 0      | 77.4 (1.14) | 46.0 (1.63) |
> | Gen 5  | 80.2 (0.86) | 46.0 (0.00) |
> | Gen 20 | 80.7 (0.90) | 49.7 (1.25) |
> | Gen 30 | 86.0 (1.56) | 55.0 (0.82) |
> | Gen 50 | 86.6 (1.03) | 60.0 (2.90) |
>
> We will specify this setting in our new version. Thank you.
>
> **W2.3** experimental design
>
> **Yes, the two downstream tasks (Boolean QA and language inference) are exactly derived from the tasks in Section 3.** We only transfer the factual question to True or False, and simplify the reasoning question by limiting it to type effectiveness. **Our purpose is to evaluate the accuracy automatically,** which is more efficient for us to perform more experiments. In the analysis experiments in Section 3, the results are evaluated by hand in most cases.
>
> **W3.1** need a detailed algorithm
>
> Thank you for your advice. We will further illustrate an algorithm of our training method to increase its clarity. Thank you.
>
> **W3.2** hyperparameter choice
>
> For each model, we train it with two learning rates 3e-4 and 8e-4. The selection of the batch size (2 and 4) is purely based on the sentence length for different tasks, to maximize the utilization of chip memories, which do not impact the eventual training performance in our experiments.
>
> **W3.3** train-test split
>
> Yes, we reserve the positive samples and resample new negative samples to be test set. **The reason is that the positive cases are more important than negative ones and we need to evaluate whether the model can better memorize them.** This is also the lurking situation of the pre-training of LLMs.
>
> **We thus report the performance on positive and negative cases separately**, as a solution for this. Please see below (under seed 42):
>
> |        | Positive | Negative |
> |--------|:-----------:|:-----------:|
> | 0      | 96.0     | 58.0     |
> | Gen 5  | 96.0     | 64.0     |
> | Gen 20 | 94.4     | 68.0     |
> | Gen 30 | 97.9     | 74.5     |
> | Gen 50 | 98.6     | 75.0     |
>
> We find that the role of self-training in classifying negative samples is more significant, while does not compromise that on positive samples.

---

> > ### Comment · Reviewer_5UvA · 2023-11-21
> > **Response + Request to see Revised Paper**
> >
> > Thank you for your detailed replies and for the extra results that you show.
> > I am very satisfied with your answers and proposed changes, but I am expecting to see the revised paper with all those changes before the end of the discussion period, if possible?
> >
> > In the meantime, I will increase my rating from 5 to 6, and I am hoping to be able to increase my rating to 7 after having verified how at least some of the changes are implemented in your revised paper.

---

> > > ### Author Response · Authors · 2023-11-22
> > >
> > > Thank you for your kind comments. We will make updates to the paper and are still working on it. Thank you for your patience.

---

> ### Author Response · Authors · 2023-11-22
>
> We have updated the revision. Please refer to the general response.
>
> Due to the time constraint, we are sorry that we cannot include the pseudocode in Sec. 4.1 during this period, which will take more time, and we also need more time to reorganize the writing. Thank you.

---

> ### Comment · Reviewer_5UvA · 2023-11-23
>
> Thank you for your efforts towards improving the paper, I am satisfied with the direction.
> I am updating my score to ~~7~~ 8 (given the lack of 7 rating value).

---

### Official Review · Reviewer_jvkM · 2023-11-04

**Soundness:** 3 good
**Presentation:** 3 good
**Contribution:** 3 good
**Rating:** 6
**Confidence:** 3

**Summary:**

The paper explores the application of  LLMs to the unique and fictional universe of the popular strategy game POKÉMON, demonstrating the outworld context can let LLMs suffer from knowledge distortions and logical flaws. It introduces POKEMON-PY, a Python library for interacting within the Pokémon world, and proposes a novel self-supervised learning method called Self-Training with Self-Competition to adapt LLMs to new world settings by enabling continuous learning through self-play. Experiments demonstrate that this method significantly improves the adaptation of the LLaMA2-7b model to perform downstream tasks within the Pokémon world.

**Strengths:**

It is interesting to discuss that the outworld context can let LLMs fail. The paper presents an interesting phenomenon that the knowledge required for reasoning greatly overlaps with answers to factual questions, but LLM performs worse in the reasoning task.

A Self-Training with Self-Competition strategy to train the LLMs to adapt to the new universe seems simple but useful, avoiding large amounts of annotated data.

**Weaknesses:**

The paper lacks comprehensive details in several areas, which are crucial for full understanding and replication:

It is significant for the author to give a specific prompt design to guide the LLM in playing Pokemon since it is well-known that LLM is sensitive to the given prompt.

Another important missed detail is for optimizing the LLM within the self-training and self-competition framework. There are many alternatives to train the LLM agent, given the collected win data.

The absence of source code and datasets further hampers the reproducibility of the results, as the paper omits many critical specifics.

**Questions:**

As mentioned in weaknesses, a deeper explanation of the training process for the LLM is necessary. For instance, the design of the loss function remains vague. Including pseudocode could greatly aid in comprehension.

Moreover, there's a concern about the utility of the LLM's outputs in game scenarios; it's unclear if the data generated during gameplay is utilized as-is, which may not reflect optimal game strategies.

I am not sure about the difficulty level of the proposed reasoning task and factual task. Can you provide more detail?

I am curious about how the LLM agent performs in real-world factual answering compared with reasoning tasks while sharing the same required knowledge base.

---

> ### Author Response · Authors · 2023-11-17
>
> Thank you for your valuable review.
>
> **W4** source code and data
>
> We will release our source code and generated data to facilitate future studies when the paper is ready to public. Thank you.
>
> **W1** specific prompt
>
> **To ensure the performance, we follow the predefined format to prompt** LLaMA2 and Alpaca. Please see below.
>
> For *LLaMA2*:
>
> [INST] Answer a given question in the Pokemon world. What is …? [/INST]
>
> For *Alpaca*:
>
> \### Instruction:
>
> Answer a given question in the Pokemon world.
>
> \### Input:
>
> What type is Pikachu?
>
> \### Response:
>
> For *ChatGPT*:
>
> Answer a given question in the Pokemon world. What type is Pikachu?
>
> **It is important to using “Answer a given question in the Pokemon world” to limit the scope of the responses, while the wording of specific questions does not matter.**
>
> **W2&Q1** detail for optimizing the LLM
>
> Thank you for your question. In our paper, **we train LLaMA with the language modeling loss (i.e. next word prediction)** on the collected win data. We show a concrete illustration of our data:
>
> *Instruction:*
>
> *Given a battle context, suggest a move for Roaring Moon.*
>
> *Input:*
> *pokemon: Roaring Moon tera: Steel moves: Dragon Dance, Earthquake, Substitute, Acrobatics opponent: Iron Hands*
>
> *Response:*
>
> *Earthquake*
>
> The model is trained to predict “Earthquake” conditioned on the previous context.
>
> We did not highlight this because there are not many options for training decoder-only models like LLaMA, while we will clarify this point in the new version, thank you.
>
> **Our training code is the same as performing supervised fine tuning (SFT) with an autoregressive LM.**
>
> We hope the concreteness above may address your concern, thank you.
>
> **Q2.1** if the collected data is utilized as-is
>
> Yes, **we do not change anything within in the data** except to make it follow the prompt format.
>
> **Q2.2** if the collected data reflects the optimal game strategy
>
> Please allow us to clarify that **our focus is to adapt the model to outworld knowledge, rather than teaching it the optimal policy to play real games.** These two objectives are not straightforwardly related. Training a good adapter is leaning to providing a nice initialization to the new world.
>
> Sure, from Table 3, we see that the trained model on Gen 50 achieves a 72.0% win rate against MaxDamage player, **which suggests that its learned policy is at least strong.**
>
> **Q3** the difficulty of the factual and reasoning tasks
>
> The factual questions we craft in the paper are very basic, e.g. types of pokemons, functions of items. All this information can be easily fetched from the web. **Hence, we think these questions are all easy for LLMs if they have seen during training.**
>
> The reasoning-required questions are on the basic selection of moves, most depending on type effectiveness. **They are also easy for humans, even the newbie in playing pokemon.**
>
> However, we find that LLMs do not work on these questions, especially the reasoning-required ones.
>
> **Q4** how LLMs perform in real-world
>
> We are sorry and not sure if we really understand your question.
>
> LLMs have a great grasp of factual questions in the real world, though there are hallucination problems in answering those questions, for example, reported in [1]. On the other hand, they are also capable of some highly complex reasoning problems, e.g. theorem proving, even difficult for humans. However, in our paper, our designed outworld questions of reasoning are very simple for humans, while LLMs show powerless.
>
> [1] Jirui Qi, et al. Cross-Lingual Consistency of Factual Knowledge in Multilingual Language Models

---

### Official Review · Reviewer_oQMe · 2023-11-06

**Soundness:** 2 fair
**Presentation:** 2 fair
**Contribution:** 1 poor
**Rating:** 5
**Confidence:** 2

**Summary:**

This paper explores the potential of Large Language Models (LLMs) to generalize beyond human knowledge, by using \textsc{Pokémon} as a case study.
Using Pokemon as a case study, an analysis is conducted on how much outworld knowledge LLMs have.
The analysis reveals that while LLMs can memorize some aspects of outworld knowledge, they often demonstrate logical inconsistencies.
Then, the author proposes a self-supervised learning method in order to adapt to a new world setting.
Experimental results show that the proposed method achieves improvements in adapting LLMs to some downstream tasks within the pokemon world.

**Strengths:**

* The paper is well-written and well-organized.
* The proposed method improves the performance of LLMs in some pokemon tasks.

**Weaknesses:**

* The motivation for choosing Pokemon as a case study of outworld is ambiguous and unclear. Even though I agree that \textsc{Pokémon} has unique and interesting worldviews, I think it is not adequate to examine LLMs' outworld knowledge through a single strategy game.
* The proposed method, which is based on a self-supervised learning method, seems to be inefficient because it learns only from the actions of the winners during self-play. Reinforcement learning algorithms such as PPO would be more efficient because they also use the loser's actions for learning models.

**Questions:**

* Why is \textsc{Pokémon} considered as a good case study for outworld?
* From the analysis of outworld knowledge in the pokemon world, what kind of behaviors can we infer that LLMs would exhibit in other new worlds in general?
* Why is the proposed method based on a self-supervised learning algorithm rather than a reinforcement learning algorithm? Is the proposed method more efficient than reinforcement learning algorithms such as PPO?

---

> ### Author Response · Authors · 2023-11-17
>
> Thank you for your valuable review.
>
> **W1&Q1** why choosing the Pokemon world as a case study
>
> **Objectively, Pokemon is well-known enough and has a complete world setting. One can find a complete database from the web to verify our statements in the paper.** In addition, it is a strategy game with clear rule settings, which suits for crafting reasoning samples.
>
> Subjectively, the research is from personal interest.
>
> We notice that it is the limitation that we cannot totally generalize the pokemon knowledge to other outworld knowledge. It is also an interesting future direction to perform a more thorough evaluation of outworld knowledge in LLMs. However, the contribution of our paper is not solely an evaluation of outworld knowledge. Thank you.
>
> **W2&Q3** compare with reinforcement learning
>
> We agree that our method only consider the samples from the winner. However, **due to the absence of loser data, our method avoids unnecessary oscillations and can faster get close to a nice initialization point**, which we seek for outworld adaption. Sure, we agree it is promising to design a ranking loss, e.g. in RLHF. We will leave this for future work.
>
> **Q2** what we can infer LLMs in other new worlds
>
> From the results in our paper, we highly conjecture that LLMs cannot tackle simple reasoning tasks in new worlds effectively, and suffer from hallucinations that are even more severe than those in the real world.

---

> > ### Comment · Reviewer_oQMe · 2023-11-23
> >
> > Thank you for your reply.
> >
> > While I appreciate the explanations provided, I still have reservations about the contributions of the paper.
> >
> > * I find it challenging to ascertain the effectiveness of the proposed method in general outworld domain scenarios based solely on its ability of outworld knowledge adaptation in Pokemon.
> > * As is known, RLHF utilizes loser data but has shown substantial performance in training LLMs. Consequently, Consequently, I'm wondering whether excluding the loser data would indeed make the learning of LLMs faster.
> >
> > Given the aforementioned points, I would like to maintain my current evaluation.

---

> > > ### Author Response · Authors · 2023-11-23
> > >
> > > Thank you for your reply.
> > >
> > > We understand your concerns and will leave them for future work.

---

### Author Response · Authors · 2023-11-17
**General Response by Authors**

We thank all reviewers all your valuable comments.

Please allow us to clarify two common concerns from the reviews:

* The training loss we use to optimize LLM is next word prediction, exactly the same as performing SFT.

* Our contributions in the paper are multi-aspect, including a Python playground for Pokemon, analysis of logical inconsistency, a new training method proposed, so it is regretful that we cannot specify all details in limited space, also including conducting a thorough evaluation of outworld knowledge from different worlds.

From the reviews, we also notice that it is the limitation in our paper that we cannot generalize the outworld knowledge in the pokemon world to other new worlds. We will leave this for future work.

Thank you.

---

> ### Author Response · Authors · 2023-11-22
> **Paper Revision**
>
> Again, thank all reviewers for the comments. We update the revision following some of the suggestions in the reviews:
>
> * related work discussion in Sec, 7;
>
> * clarification of the training loss and training setting in Sec. 4, and update of Table 5 results for multiple runs;
>
> * complemented definition of the outworld knowledge in Sec. 1;
>
> * experiment for the selection of $\epsilon$ in Appendix.
>
> We promise that we will follow all suggestions in the reviews to further improve our paper because they are really helpful, while due to the time constraint, we couldn't manage to fill in all the details in the current revision in the discussion period.

---

### Meta-Review · Area_Chair_Fd6N · 2023-12-08

**Metareview:**

This paper introduces the concept of outworld knowledge, proposes Pokemon as a benchmark and tests a range of well established pre-trained LLMs. The topic of benchmark tasks for LLM agents is very timely and likely to be of broad interest to the ICLR community. However, the paper currently has issues with clarity and would benefit significantly from a second benchmark task to ensure the conclusions reached apply beyond Pokemon.

**Justification For Why Not Higher Score:**

+ A second environment would significantly improve confidence in the empirical results presented

**Justification For Why Not Lower Score:**

N/A

---

### Decision · Program_Chairs · 2024-01-16

Reject